# Transmission competence of a new mesonivirus, Yichang virus, in mosquitoes and its interference with representative flaviviruses

Guoguo Ye[1,2], Yujuan Wang[3], Xiaoyun Liu[4], Qiannan Dong[1,2], Quanxin Cai[1], Zhiming Yuan[1,2]*, Han Xia[1,2]*

**1** Key Laboratory of Special Pathogens and Biosafety, Wuhan Institute of Virology, Center for Biosafety Mega-Science, Chinese Academy of Sciences, Wuhan, China, **2** University of Chinese Academy of Sciences, Beijing, China, **3** Shanghai Public Health Clinical Center, Fudan University, Shanghai, China, **4** Shandong Provincial Collaborative Innovation Center for Antiviral Traditional Chinese Medicine, Shandong University of Traditional Chinese Medicine, Jinan, China

* yzm@wh.iov.cn (ZY); hanxia@wh.iov.cn (HX)

**Data Availability Statement:** All relevant data are within the manuscript and its Supporting Information files.

## Abstract

Advances in technology have greatly stimulated the understanding of insect-specific viruses (ISVs). Unfortunately, most of these findings are based on sequencing technology, and laboratory data are scarce on the transmission dynamics of ISVs in nature and the potential effects of these viruses on arboviruses. Mesonivirus is a class of ISVs with a wide geographical distribution. Recently, our laboratory reported the isolation of a novel strain of mesonivirus, Yichang virus (YCV), from *Culex* mosquitoes, China. In this study, the experimental infection of YCV by the oral route for adult and larvae mosquitoes, and the vertical transmission has been conducted, which suggests that YCV could adopt a mixed-mode transmission. Controlled experiments showed that the infectivity of YCV depends on the mosquito species, virus dose, and infection route. The proliferation curve and tissue distribution of YCV in *Cx. quinquefasciatus* and *Ae. albopictus* showed that YCV is more susceptible to *Ae. albopictus* and is located in the midgut. Furthermore, we also assessed the interference of YCV with flaviviruses both *in vitro* and *in vivo*. YCV significantly inhibited the proliferation of DENV-2 and ZIKV, in cell culture, and reduced transmission rate of DENV-2 in *Ae. albopictus*. Our work provides insights into the transmission of ISVs in different mosquito species during ontogeny and their potential ability to interact with mosquito-borne viruses.

## Author summary

Mosquitoes transmit many pathogenic viruses, such as dengue virus, Zika virus, and Japanese encephalitis virus, which are of great burden to public health worldwide, notably in the tropical regions. In addition, they also harbor a number of insect-specific viruses (ISVs), however, little is known about the role for the ISV in mosquito populations and their interaction with arboviruses. Yichang virus (YCV) is a newly identified member of

**Funding:** ZY received awards of 2018ZX10101004 and 2013FY113500 from the Ministry of Science and Technology of the People's Republic of China; HX received the award of WIV-135-PY2 from the Wuhan Institute of Virology, China. The funders had no role in study design, data collection and analysis, decision to publish, or preparation of the manuscript.

**Competing interests:** The authors have declared that no competing interests exist.

the *Mesoniviridae* family discovered in *Culex* mosquitoes collected from Hubei, China. In this article, we investigate the transmission competence of *Cx. quinquefasciatus* and *Ae. albopictus* for YCV, and its interaction with mosquito-borne flaviviruses *in vitro*. Our study shows, for the first time, that this virus could be horizontally and vertically transmitted, thus strongly inhibiting the replication of the pathogenic dengue virus serotype 2 (DENV-2)in C6/36 cells, and reduced transmission of DENV-2 in *Ae. albopictus*. These data provide further insight on the transmission mechanism of ISVs and how they interact with flaviviruses.

## Introduction

For the last two decades, with advancements in high-throughput sequencing, metagenomics and intensified mosquito surveillance, a large number of insect-specific viruses (ISVs) and arboviruses have been discovered. ISVs are restricted to arthropods and are unable to replicate in vertebral cells [1], whereas arboviruses can be transmitted between mosquitoes and vertebrates [2]. Although these two types of viruses have different host ranges, there are many similarities in virological classification and transmission methods [2]. While arboviruses are related to many diseases of animals and humans, such as dengue virus (DENV), Zika virus (ZIKV), and Japanese encephalitis virus (JEV), their horizontal transmission and vertical transmission mechanisms are relatively clear [3–8]. Significantly, the mode of transmission and ecological significance of the ISVs remain unknown 45 years after the discovery of ISVs by Stellar and Thomas [9,10].

Most ISVs and arboviruses share similar genetic and structural virus particles and have a close evolutionary relationship. The relationship and interaction of the ISVs and arboviruses are complex and attract concerns worldwide. Some scholars conjecture that ISVs may be an important evolutionary source of new arboviruses [11–13] and that some ISVs may inhibit arbovirus infections in their insect hosts [10]. Moreover, the host-restricted characteristics of ISVs facilitate their development and application in biologically controlling the spread of infectious viruses [14]. In general, research on ISVs transmission models and interactions with arboviruses is becoming increasingly important.

*Mesoniviridae* contains single-stranded positive-sense RNA viruses belonging to *Nidovirales*. So far, all members of the *Mesoniviridae* family are ISVs, with a wide geographic and population distribution [15]. Among them, Yichang virus (YCV), isolated from *Culex* mosquitoes collected from Hubei, China, has the largest genome ever discovered [16]. Due to the extensive geographic distribution and host range of *Mesoniviridae*, as well as their potential as biological control agents, more studies are needed to better understand arthropod-restricted virus maintenance in nature and the potential impact on arbovirus infection.

In this study, we investigated horizontal transmission through an oral virus challenge at the adult stage and feeding with virus-contaminated water during the larval stage and vertical transmission of YCV in *Culex quinquefasciatus* and *Aedes albopictus*. The effects of virus titers, breeding water and mosquito species on YCV infectivity were also confirmed. Furthermore, we analyzed the proliferation and tissue distribution of YCV in mosquitoes and their interaction with flavivirus *in vitro* and *in vivo*. Our findings deepen the understanding of the transmission mechanism of ISVs and provide important information for the implementation of ISVs in vector-borne virus control.

## Materials and methods

### Mosquito rearing

Both *Cx. quinquefasciatus* and *Ae. albopictus* (kindly provided by Chinese center for disease control and prevention, Beijing, China) were maintained at 27±1˚C with a 12:12 light: dark cycle and 70% relative humidity. Adult mosquitoes were provided with 10% glucose solution. Defibrillated horse blood (Shanghai Yuanye Biological Technology Co., Ltd. Shanghai, China) was provided through the Hemotek membrane blood feeding system (Hemotek, Lancaster, UK). F1 generation of mosquitoes used were collected from parent females oviposited after the blood meal.

### Virus strain

The YCV (strain HB14-64-01) used in this study was originally isolated in a culture of C6/36 cells inoculated with homogenized *Culex* mosquitoes from the Yichang area of Hubei [16]. C6/36 cells were cultured as monolayers in T75 flasks at 28˚C in RPMI medium (Gibco, Carlsbad, USA) supplemented with 10% FBS, 2% tryptose phosphate broth (Gibco); YCV was grown in C6/36 cells for viral production. DENV-2 (strain TSV01), ZIKV strain (SZ-WIV01) and JEV (strain SA14) were kindly provided by Professor Bo Zhang (Wuhan Institute of Virology, Chinese Academy of Sciences, Wuhan, China).

### Plaque assay

BHK-21 cells in 24-well plates were infected with 10-fold serial dilutions of viruses for 1 h at 37˚C. The cell monolayers were overlaid with 1% Aquacide II (Calbiochem, Saint Louis, USA) in DMEM containing 2% FBS and incubated at 37˚C for 4 days. Monolayer cells were fixed with 3.7% formaldehyde and stained with 1% crystal violet to visualize plaques.

### Plasmid construction and antibodies

The cDNA encoding the full-length YCV nucleotide capsid protein (abbreviation YCV-N) was coded by ORF2b cloned into vector pet28a using primers (YCV Forward-1: 5′-atgccaggacgcac-caacaca-3′ and YCV Reverse-1: 5′-ggggtcaacagtaataacataatcagcag-3′). Recombinant protein expression was induced in the *E. coli* BL21 DE3 strain using 1 M IPTG for inclusion bodies, separated, and purified by SDS-PAGE. The protein was used for subsequent polyclonal antibody preparation in the laboratory at the Animal Center, Wuhan Institute of Virology, Chinese Academy of Sciences, following standard animal procedures. A 178-bp fragment between nucleotides 154 and 331 of YCV was amplified by specific primers (YCV Forward-2: 5′-ccaggtttgagcgaacaggt-3′; and YCV Reverse-2: 5′-tcggggtgcggttaaaagtg-3′) and cloned into the pet28a vector for constructing the standard. A 127-bp fragment between nucleotides 10517 and 10643 of DENV-2 (TSV01) was amplified by specific primers (Forward-2: 5′- tccctta-caaatcgcagcaac-3′; and Reverse-2: 5′- tggtctttcccagcgtcaat-3′) and cloned into the pet28a vector for constructing the standard.

### Mosquito infection by blood meal

Before blood meal, 5 to 7-day-old female mosquitoes were starved for at least 16 h. The mosquitoes were fed defibrillated horse blood (Shanghai Yuanye Biological Technology Co., Ltd. Shanghai, China) mixed with YCV solution at 1:1 via the Hemotek membrane blood feeding system (Hemotek, Lancaster, UK). Mosquitoes were allowed to feed for 1 h in light conditions, at 24˚C and 70% relative humidity (RH). Fully engorged mosquitoes were selected and incubated at 28˚C and 70% relative humidity (RH) with 10% glucose solution, and around 10

mosquitoes were collected at 0, 3, 7, 11, and 14 days post infection (dpi) for viral detection each time and with three repeats.

## Mosquito infection by breeding in YCV-containing liquid

Water (clean water and sewage from NO. 52 Hongshan Side Road, Wuhan, China) (S1 Table) was mixed with YCV supernatant from infected C6/36 cells. In addition, sewage water was filtered at 0.22 μm to remove particles, clean water together with addition of particles derived from sewage were used for mosquito breeding. The initial YCV titer were $1 \times 10^5$, $1 \times 10^4$, $1 \times 10^3$ or $1 \times 10^2$ pfu/ml. The mixture was used for breeding the 3–4 instar larvae of *Cx. quinquefasciatus* and *Ae. albopictus*. After a 1–2 d exposure to the mixture, the larvae were transferred to a container with fresh water until eclosion. The emerging mosquitoes were reared for an additional 8 days for viral detection. Sample size ranged between 50–70 immature mosquitoes each time, and with three repeats.

## Viral RNA detection in mosquito

Total RNA was isolated using RNAiso Plus reagent (Takara, Dalian, Japan) according to the manufacturer's protocol and dissolved in 60 μL RNase-free water. Real-time PCR was performed using the One Step SYBR Prime Script PLUS RT-PCR Kit (Takara) on a MyiQ Optics Module (Bio-Rad, Hercules, USA). The primers were as follows: YCV detection primers are YCV Forward-2 and YCV Reverse-2 in Plasmid construction and antibodies section. AalRPS17 was used as the internal control for qRT-PCR. The primer sequences for AalRPS17 primers were AalRPS17 forward: 5′- acgtagttgtctctctgcgctc-3′ and AalRPS17 reverse: 5′-cgcttggtttcgtgacacatc-3′[17]. A standard curve of YCV (linear curve slope –3.5715, Y intercept 41.552, $R2 = 0.9988$, amplification efficiency 90.546) and DENV-2 (linear curve slope –3.5841, Y intercept 45.146, $R2 = 0.9998$, amplification efficiency 90.543) was generated from a range of serial 10-fold dilutions of the plasmid and was used to normalize the genomic copies.

## Western blotting

At 14 dpi, heads, midguts and whole bodies of 5–10 mosquitoes were washed three times with cold PBS. Total protein was extracted by homogenizing samples and lysing in RIPA buffer (Sangon, Shanghai, China) for 30 min at 4°C. Protein concentration was determined using the BSA protein Assay Kit (Takara). YCV was detected with a mouse polyclonal antibody produced by immunizing mice with protein YCV-N. Actin was used as an internal control and detected with a mouse pan-actin antibody-clone C4 (Millipore, Billerica, MA).

## *Co-infection interference* in vitro

**Simultaneous infections:** Before infection, YCV was mixed with DENV-2, ZIKV or JEV according to the specified multiplicity of infection (MOI) (YCV MOI = 1, DENV-2/ZIKV / JEV MOI = 1 or 0.1; YCV MOI = 0.1, DENV-2/ZIKV / JEV MOI = 1 or 0.1) and then incubated with C6/36 or Aag2 cells for 1 h at 28°C with 5% $CO_2$.

 **Sequential infections:** The method of sequential infections was similar to that for simultaneous infection, but YCV (MOI = 1) was incubated with C6/36 after 12 h, then the C6/36 cells containing YCV were incubated with DENV-2, ZIKV or JEV. After incubation, cells were rinsed once with PBS and 1 mL of complete media was added. Cells were incubated for 5 days at 28°C with 5% $CO_2$.

 Simultaneous and sequential infections supernatant was collected at 1, 2, 3, 4 and 5 dpi for the plaque assay to determine infectious viral particle. The data were repeated three times.

### *Co-infection interference* in vivo

Mosquitoes that pulled through 5–7 d after being starved for at least 16 h were fed defibrillated horse blood (Shanghai Yuanye Biological Technology Co., Ltd) mixed with YCV solution and DENV-2 solution at 1:1:1 using the Hemotek membrane blood feeding system (Hemotek). The titer for YCV and DENV-2 used was both $3.7 \times 10^7$ pfu/ml. Full engorged mosquitoes were selected and incubated at 28˚C and 70% relative humidity (RH), and the bodies and heads of 10 to mosquitoes were collected at 7 and 14 dpi for viral detection. The data were repeated by three times.

Evaluation of *Ae. albopictus* vector competence for DENV-2 used infection rate and population transmission rate, as follows[18]:

Infection rate: Percentage of mosquitoes containing virus in their bodies (number positive/number tested)

Transmission rate: Percentage of mosquitoes containing virus in their heads (number positive/number tested)

### Statistical analysis

Statistical analysis (Student's t-test) was performed using R.3.5.1 (https://www.r-project.org/). In all tests, the data represent the mean ± SEM. The results were analyzed using the unpaired t-test with statistical significance alpha (α) levels denoted as P < 0.05, *; P < 0.001, ** and P < 0.001, ***.

## Results

### Mosquitoes are permissive to oral YCV infection in a dose-dependent manner

Research on the route of YCV oral infection showed that YCV reproduction was detected in total RNA from the whole mosquito bodies of *Cx. quinquefasciatus* (26.9%) and *Ae. albopictus* (45.5%) by RT-PCR (Table 1 and S1 Fig). Western blot results demonstrated that the YCV structural protein N was expressed in both *Cx. quinquefasciatus* and *Ae. albopictus* (S1 Fig). We also investigated the infectious activity of these viruses in C6/36 cells. In comparison to the control, 17.6% and 33.3% of the homogenates of *Cx. quinquefasciatus* and *Ae. albopictus* resulted in cytopathy and positive for YCV RNA in C6/36 cells, respectively (Table 1 and S1 Fig). All the results indicate that YCV can infect two types of adult mosquito through the oral route.

The infectivity effect factor of the virus dose was verified with a series of YCV titers (from $1 \times 10^2$ to $1 \times 10^6$ pfu/ml) to infect mosquitoes. YCV infected both species of mosquito at $1 \times 10^6$ pfu/ml by blood-meal feeding, with a stronger infectivity in *Ae. albopictus*. The average copies (log10) and the infection rate of YCV in *Ae. albopictus* were higher than in *Cx. quinquefasciatus* at $10^6$ pfu/ml (Fig 1A and 1B). However, with lower titers of YCV, its infectivity to both mosquito species decreased dramatically. The $1 \times 10^5$ pfu/ml virus infection resulted in 4.94 and 4.5 average copies (log10) of YCV in *Ae. albopictus* and *Cx. quinquefasciatus* (Fig 1A),

**Table 1. Infection rate of YCV in *Cx. quinquefasciatus* and *Ae. albopictus*.**

| Species | Blood meal titer (pfu/ml) | Infection rate | |
|---|---|---|---|
| | | No. YCV RNA positive/No. engorged | No. CPE observed/No. engorged |
| *Cx. quinquefasciatus* | $10^6$ | 26.9% (7/26) | 3/17 (17.6%) |
| *Ae. albopictus* | $10^6$ | 10/22 (45.5%) | 4/12 (33.3%) |

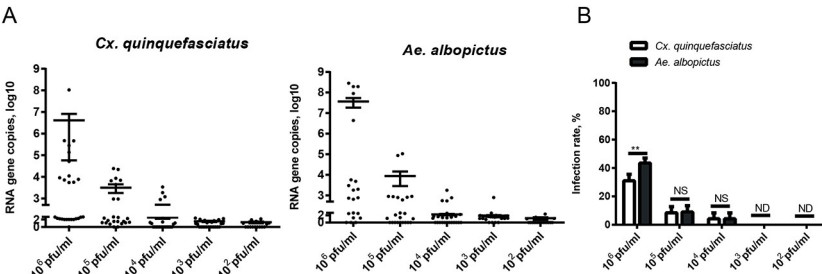

**Fig 1. Infectivity and infection rates of *Cx. quinquefasciatus* and *Ae. albopictus* infected with different doses of YCV.** Mosquitoes were inoculated with an infectious blood meal containing a dose of $10^6$, $10^5$, $10^4$, $10^3$ or $10^2$ pfu/ml YCV. (A) Ingested virus titers of *Cx. quinquefasciatus* and *Ae. albopictus* by qRT- PCR at 14 dpi. Each dot represents one mosquito body; the results are expressed as the mean ± SEM. (B) Infection rates of *Cx. quinquefasciatus* and *Ae. albopictus* at 14 dpi presented as the percentage of the total number of engorged mosquitoes. The results are expressed as the mean ± SEM. The results were analyzed using the unpaired t-test. A P value of < 0.05 indicates statistical significance. P < 0.05, *; P < 0.001, ** and P < 0.001, ***.

corresponding to 8.9% and 8.5% infection rates, respectively (Fig 1B). When the infective YCV titer was less than $1\times10^4$ pfu/ml, no infection was observed in the two mosquitoes (Fig 1A and 1B).

## Mosquitoes could acquire YCV in an aquatic environment, and particles enhance virus acquisition ability

The investigation of the potentiality of the virus to infect larvae (S1 Table) indicates that the average copies (log10) of YCV in *Cx. quinquefasciatus* were not significantly different when mosquitoes were bred in sewage (5.13) or clean water (5.17) containing $10^5$ pfu/ml of YCV (Fig 2A), while the positive rate in sewage (10.66%) was slightly higher than that in clean water (5.66%) (Fig 2B). Interestingly, at the treatment concentrations of $10^3$ pfu/ml and $10^4$ pfu/ml, the virus average copies (log10) were much higher for *Cx. quinquefasciatus* (4.04 and 4.4.65) in the sewage than in clean water (1.2 and 3.62) (Fig 2A), but the positive rate was low (<3%) (Fig 2B). Similarly, the infection pattern of YCV on *Ae. albopictus* was comparable to that of *Cx. quinquefasciatus* in the sewage and clean water (Fig 2A and 2B). The results show that the infectivity present in larvae incubated in clean water or sewage was dose-dependent; *Ae. albopictus* was more susceptible to YCV; and YCV had a stronger infection ability in sewage for both two mosquito species.

In addition, the average copies increased by approximately $10^2$-fold in the clean water with added particles, with a significant increase in infection rate from 2.17% to 13.2% in *Cx. quinquefasciatus* (Fig 3A and 3B). In contrast, the average copies of YCV in *Cx. quinquefasciatus* reduced by approximately $10^4$-fold with a lower infection rate of 1.43% in the filtered sewage compared to the sewage treatment. Furthermore, along with the restoration of the particles in sewage, the average YCV copy number in *Ae. albopictus* increased by 10-fold, and the infection rate conclusively increased from 16.05% to 39.68%. However, the average YCV copy number in *Ae. albopictus* decreased by approximately $10^2$-fold when incubated in filtered sewage compared to clean water, and the infection rate also decreased from 34.65% to 15.04% (Fig 3A and 3B). This result wholly suggests that the particles in water influence the infectivity of YCV to mosquitoes, which might be related to the viral stability, but remains to be further investigated.

## YCV vertical transmission in mosquitoes

The presence of YCV in different developmental stages of the F1 generation detected showed that YCV copies were low among the eggs, larvae, pupae and adults of *Cx. quinquefasciatus*

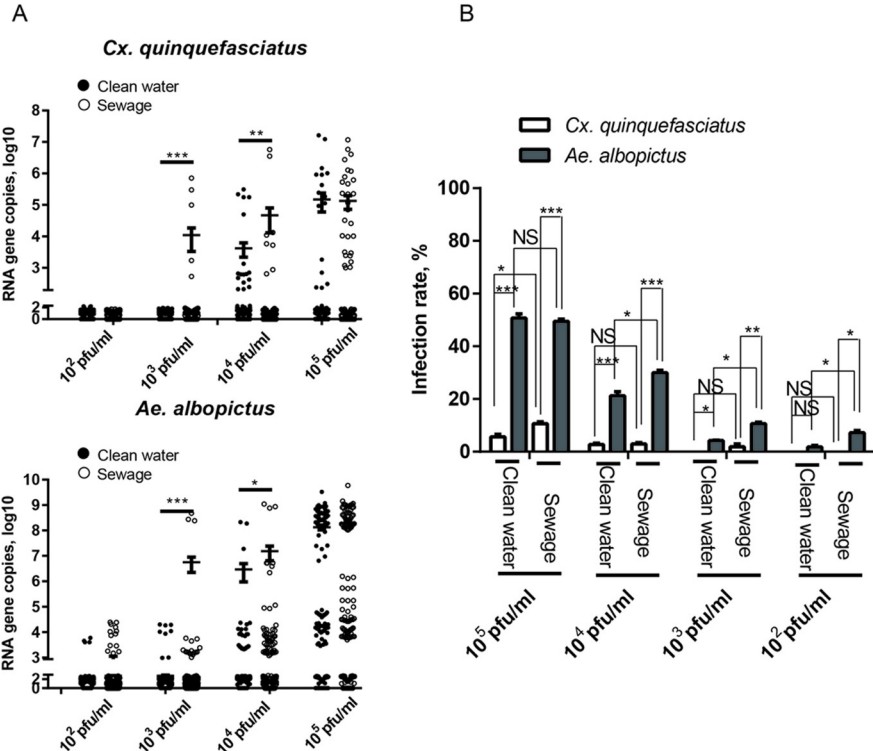

**Fig 2. YCV in an aquatic environment can be acquired by *Cx. quinquefasciatus* and *Ae. albopictus*.** *Cx. quinquefasciatus* and *Ae. albopictus* were bred in clean water or sewage with a serial YCV titration ($10^5$, $10^4$, $10^3$ and $10^2$ pfu/ml). (A) The emerging adults were reared for an additional 8 days for YCV detection by qRT-PCR. Each dot represents one mosquito body; the results are expressed as the mean ± SEM. (B) Infection rates of *Cx. quinquefasciatus* and *Ae. albopictus* are represented as the ratios of mosquito infection. The results are expressed as the mean ± SEM. The results were analyzed using the unpaired t-test. A P value of < 0.05 indicates statistical significance. P < 0.05, *; P < 0.001, ** and P < 0.001, ***.

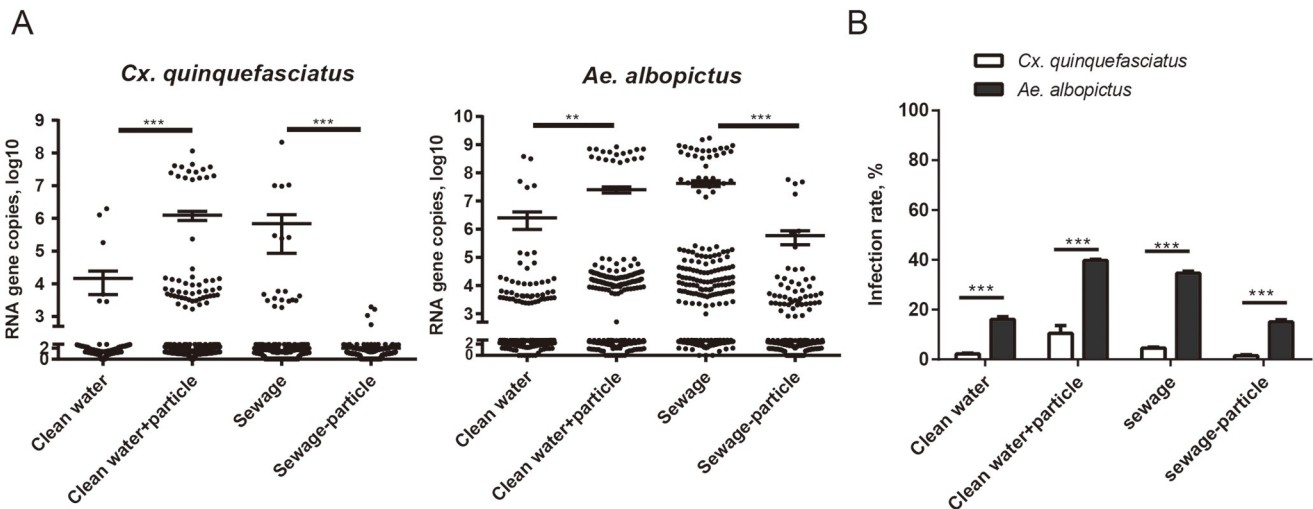

**Fig 3. Particulate matter in sewage enhances YCV infectivity of *Cx. quinquefasciatus* and *Ae. albopictus*.** Larvae (3rd-4th instar) were bred in clean water, clean water+particles, sewage, or sewage-particles with YCV (final YCV titer was $10^4$ pfu/ml). (A) YCV RNA titers of *Cx. quinquefasciatus* (left) and *Ae. albopictus* (right) were detected by qRT-PCR 8 days after emergence. One dot represents one mosquito. The results are expressed as the mean ± SEM. (B) Infection rates. The results were analyzed using the unpaired t-test. A P value of < 0.05 indicates statistical significance. P < 0.05, *; P < 0.001, ** and P < 0.001, ***.

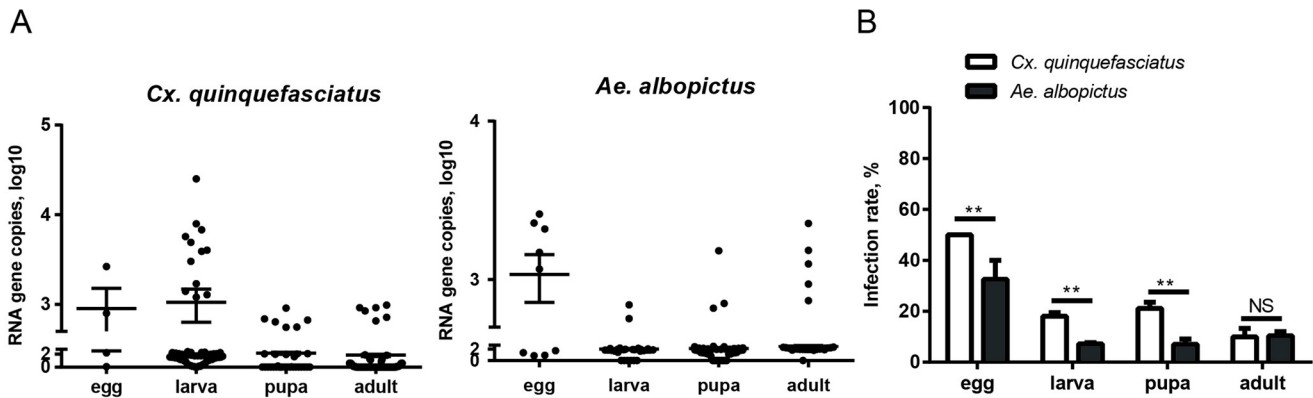

**Fig 4. Viral load of YCV in different life stages of the F1 generation.** After the infected *Cx. quinquefasciatus* and *Ae. albopictus* parent females had oviposited, egg papers were hatched in deoxygenated water. (A) Offspring were reared and YCV RNA copies were detected by qRT-PCR in different life stages (egg, larva, pupa and adult). One dot represents one mosquito. The results are expressed as the mean ± SEM. (B) Positive rate. The results are expressed as the mean ± SEM. The results were analyzed using the unpaired t-test. A P value of < 0.05 indicates statistical significance. P < 0.05, *; P < 0.001, ** and P < 0.001, ***.

(Fig 4A). Adults had the lowest positive rate (10%), quantifiably, there were no marked differences between larvae (18.05%) and pupae (21.13%) (Fig 4B). As such, higher virus copies and positivity rates were detected in adults of the *Ae. albopictus* F1 generation than in larvae and pupae (Fig 4A and 4B). Although vertical transmission is the primary way to maintain and transmit ISVs in nature, the lower vertical transmission efficiency of YCV suggests that vertical transmission may only be one of its transmission methods.

## Proliferation and distribution of YCV in mosquitoes

RT-qPCR revealed that all *Ae. albopictus* were YCV virus positive at 0 dpi (Fig 5B); the YCV copies increased steadily until 7 dpi and then maintained a stable level in later periods (p>0.05) (Fig 5A). On the other hand, the virus average copies (log10) in *Cx. quinquefasciatus* steadily decreased in the first three days, but increased to 6.67 at 7 dpi, and then maintained at

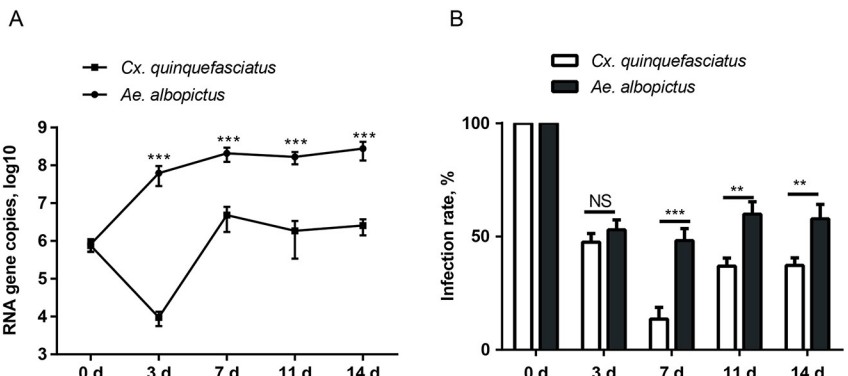

**Fig 5. Virus reproduction and infection rates for YCV in *Cx. quinquefasciatus* and *Ae. albopictus*.** The mosquitoes were fed a mixture containing horse blood (50% v/v) with 10^6 pfu/ml YCV. (A) YCV virus RNA copies in the whole mosquito bodies were detected by qRT- PCR at 0 dpi, 3 dpi, 7 dpi, 11 dpi, and 14 dpi. The results are expressed as the mean ± SEM. (B) Infection rates of *Cx. quinquefasciatus* and *Ae. albopictus* at the indicated time points presented as the percentage of the total number of engorged mosquitoes. Shown are the mean percentages from three independent replicates. Error bars show the standard error of the mean. The results were analyzed using the unpaired t-test. A P value of < 0.05 indicates statistical significance. P < 0.05, *; P < 0.001, ** and P < 0.001, ***.

the same level in later (p>0.05) (Fig 5A). It is clear that not only the higher infection rate but also higher viral copies at dpi 4, 7, 10, and 14, were observed in *Ae. albopictus*, suggesting that *Ae. albopictus* may be a better host for YCV than *Cx. quinquefasciatus*.

Furthermore, Western blot results showed that the YCV structural protein N has been detected in the midgut of *Cx. quinquefasciatus* and *Ae. albopictus*, but not in the head (Fig 6A). YCV nucleic acid was detected in the midgut, with higher RNA expression in *Ae. albopictus* than in *Cx. quinquefasciatus*, but not in the head of both mosquitoes (Fig 6B).

### YCV inhibits replication of pathogenic flaviviruses in vitro

As shown in Fig 7, the presence of YCV significantly inhibits DENV-2 replication in both C6/36 and Aag2 cells, with replications of about $10^3$, and $10^2$ fold lower at 4 or 5 dpi when DENV-2 is co-inoculated with YCV (Fig 7A and S2A Fig) in both cells. In addition, a 10-fold lower ZIKV replication was observed when ZIKV were co-inoculated with YCV than ZIKV alone in the C6/36 cells at 3, 4, and 5 dpi (Fig 7B and S2B Fig). However, no synergism or inhibition of YCV on JEV replication in the two cell lines was observed for all tested co-inoculation combinations (Fig 7C and S2C Fig). Furthermore, the effects of the infection sequence order were also tested, and the replication of DENV-2 and ZIKV is $10^4$ and 10-fold lower, respectively, in C6/36 cells pre-infected with YCV for 12 h in comparison with the control, whether the MOI of DENV-2 and ZIKV was high or low (Fig 8A and 8B, S3A and S3B Fig).

### YCV interferes with transmission of DENV-2 in vivo

At 7 dpi, there was no significant difference in the positive rate and viral copies of DENV-2 in the mosquito bodies of DENV-2 and DENV-2+YCV infected groups (P > 0.05) (Table 2 and

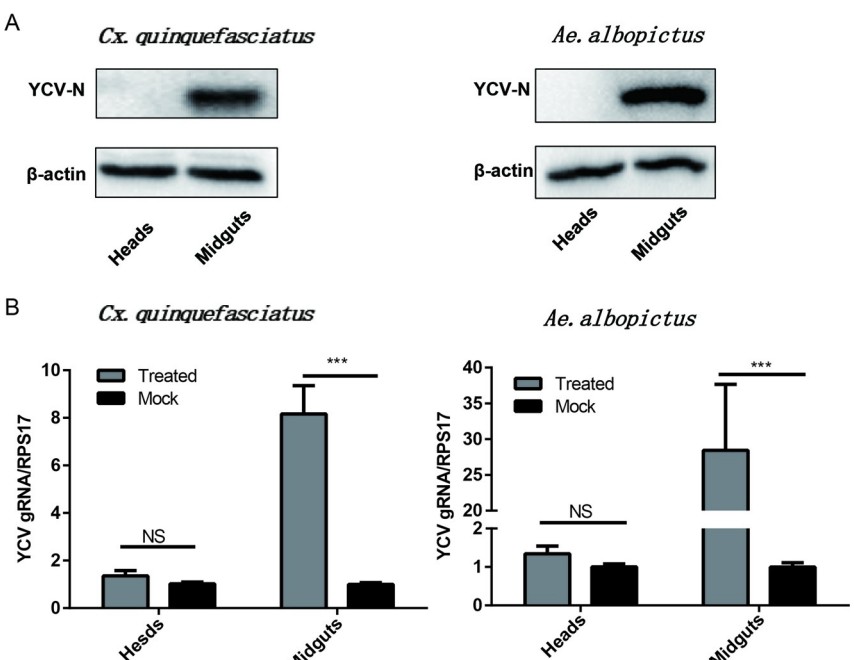

**Fig 6. Assessing the infectivity of midgut and head of *Cx. quinquefasciatus* and *Ae. albopictus*.** The mosquitoes were fed a mixture containing horse blood (50% v/v) with $10^6$ pfu/ml YCV titration. Viral protein and RNA of the midgut and head of *Cx. quinquefasciatus* and *Ae. albopictus* were detected by Western blot (A) and quantitative reverse transcription PCR (B) at 14 dpi. Data are presented as the mean ± SEM. The results were analyzed using the unpaired t-test. A P value of < 0.05 indicates statistical significance. P < 0.05, *; P < 0.001, ** and P < 0.001, ***.

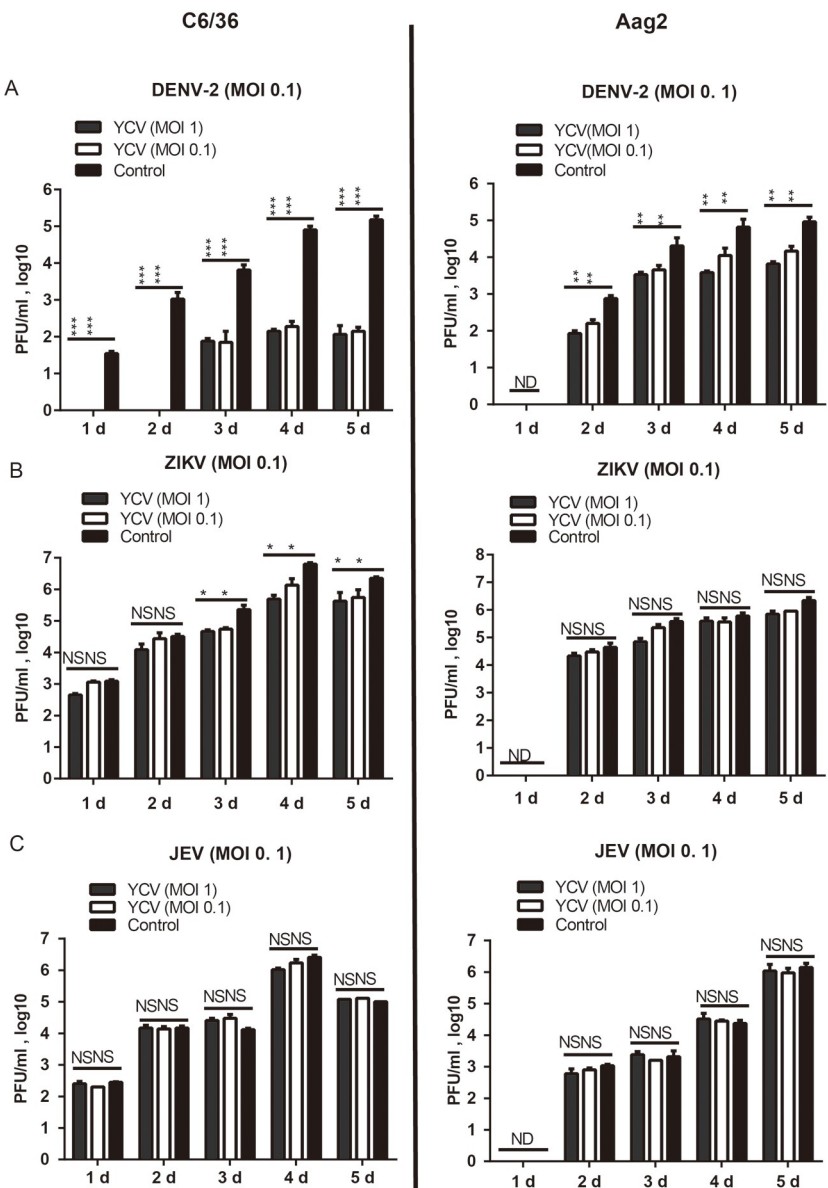

**Fig 7. YCV effects the growth of DENV-2, ZIKV and JEV in C6/36 and Aag2 cells during single- and coinfections of simultaneous infections.** The flaviviruses DENV-2, ZIKV and JEV at MOI = 0.1 were mixed with YCV at MOI = 1 or 0.1; the mixture was then added to C6/36 (left) or Aag2 (right) cells. (A, B, C) The virus titers during single- and coinfection were determined by plaque assays at the indicated time points. Data are presented as the mean of three independent experiments ±SEM. The results were analyzed using the unpaired t-test. A P value of < 0.05 indicates statistical significance. P < 0.05, *; P < 0.001, ** and P < 0.001, ***.

Fig 9). As much as significant difference was observed in transmission rate of DENV-2 between these two groups (DENV-2 (36.4%) vs DENV-2+YCV (21.1%)) (P < 0.001) (Table 2), and RNA copies of DENV-2 detected in the head of DENV-2 group were significantly higher than co-infection group (P < 0.001) (Fig 9). Similarly, at 14 dpi, the RNA copies of DENV-2 in bodies was insignificantly different between the two groups (P > 0.05) (Fig 9). However, the infection rate (90.5% vs 62.5%, p < 0.001), average virus copies (log10) in the mosquito head

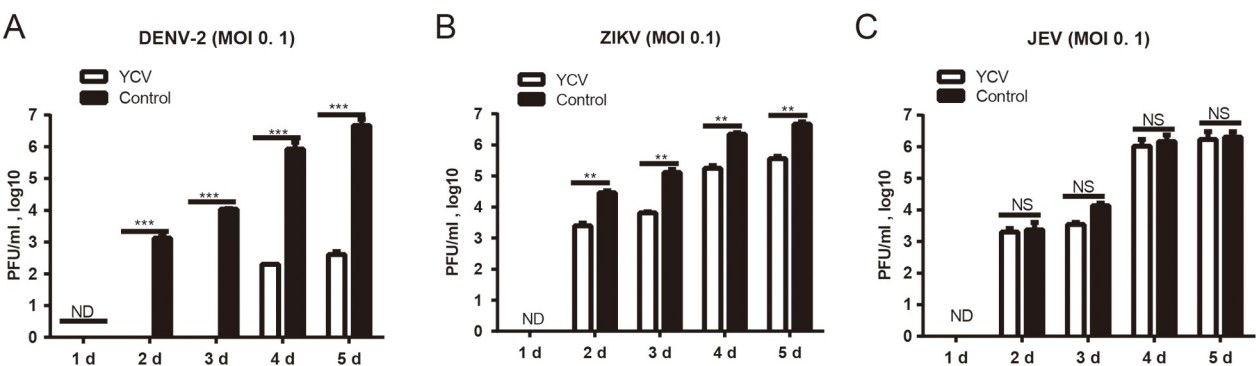

**Fig 8. YCV effects the growth of DENV-2, ZIKV and JEV in C6/36 cells during single- and coinfections of sequential infections.** After 12 h of YCV (MOI = 1) infection, C6/36 cells were infected with the flaviviruses DENV-2, ZIKV or JEV at MOI = 0.1. (A, B, C) The virus titers during single- and coinfection were determined by plaque assays at the indicated time points. Data are presented as the mean of three independent experiments ±SEM. The results were analyzed using the unpaired t-test. A P value of < 0.05 indicates statistical significance. P < 0.05, *; P < 0.001, ** and P < 0.001, ***.

(5.66 vs 5.18, p = 0.0311), and the transmission rate (71.4% vs 37.5%, p < 0.001) of DENV-2 in single infection with DENV-2 was much higher than the co-infection group with YCV and DENV-2 (Table 2 and Fig 9).

## Discussion

Herein the results presented indicate that the YCV may be transmitted and maintained in the environment using a complex transmission and maintenance model, including virus infection and transmission in the larval and adult stages, as well as localization in mosquito tissues [19]. The localization of YCV in midgut tissues was consistent with other insect-specific flavivirus (ISFs), which appeared unable to break through the barrier between the midgut and salivary glands and dissemination into the salivary glands [20–22]. Although YCV can effectively infect *Cx. quinque-fasciatus* and *Ae. albopictus* at $10^6$ pfu/ml by the oral route, its infection ability is significantly reduced and actually vanishes with reducing titers, suggesting that oral susceptibility can vary greatly and only high titers of YCV can infect *Cx. quinquefasciatus* and *Ae. albopictus*, which are consistent with that reported by Vasilakis, N. and Nasar, F [23,24]. However, recent studies using *Dianke virus*, (DKV) showed conflicting results. *Culex quinquefasciatus*, *Cx. tritaeniorhynchus*, and *Ae. aegypti* were highly susceptible to infection and able to transmit DKV [25]. This may be dependent on mosquito species and the microorganisms carried by the mosquito itself.

We conducted growth experiments with YCV in *Cx. quinquefasciatus* and *Ae. albopictus*. The results revealed that the virus infects and replicates in the midgut and that its infection rate is much lower in *Cx. quinquefasciatus* than in *Ae. albopictus*. This indicates that the

**Table 2. Infection and transmission rate of DENV-2 in single infection or co-infection with YCV in *Ae. albopictus*.**

| Group | Days post exposure | Infection rate[a] | Transmission rate[b] |
|---|---|---|---|
| DENV-2 | 7d | 50% (11/22) | 36.4%(8/22) |
| | 14d | 90.5%(19/21) | 71.4%(15/21) |
| DENV-2 and YCV co-infection | 7d | 57.9%(11/19) | 21.1%(4/19) |
| | 14d | 62.5%(10/16) | 37.5%(6/16) |

a. Percentage of mosquitoes containing virus in their bodies (number positive/number tested)

b. Percentage of mosquitoes containing virus in their heads (number positive/number tested)[18]

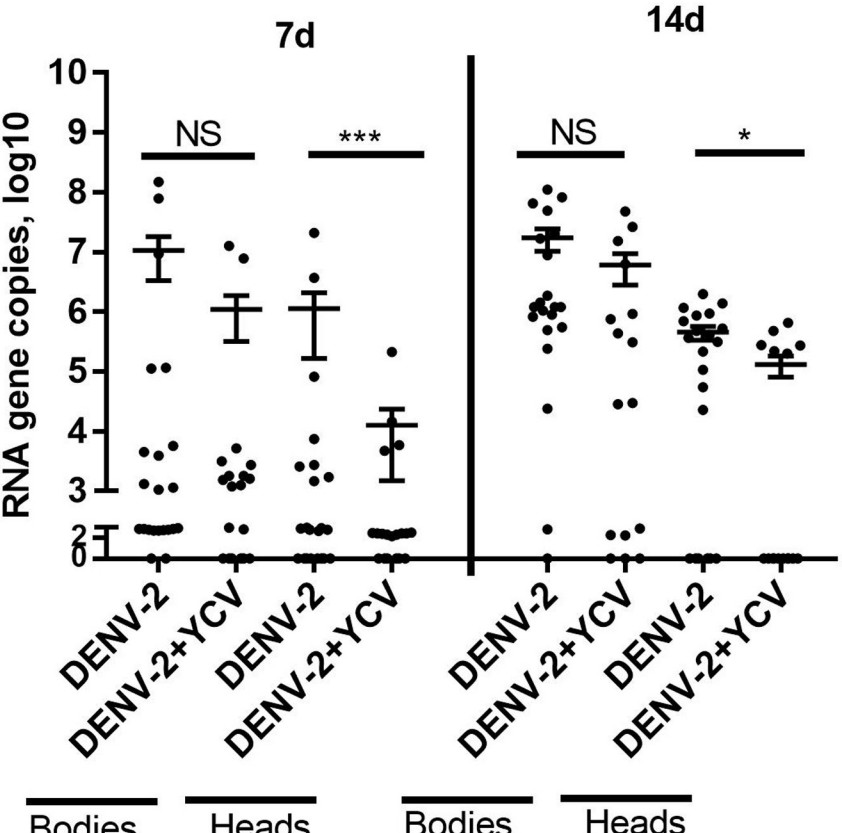

**Fig 9. Co-infection interference in vivo.** DENV-2 copies in the bodies and heads of *Ae. albopictus* DENV-2 single infection or DENV-2+YCV co-infection were detected by qRT- PCR at 7 dpi, and 14 dpi. The results for viral copeies are expressed as the mean ± SEM. The results were analyzed using the unpaired t-test. A P value of < 0.05 indicates statistical significance. P < 0.05, *; P < 0.001, ** and P < 0.001, ***.

midgut may be the main target organ for the infection and multiplication of YCV, and *Ae. albopictus* may be a more susceptible host for YCV infection, similar to what was observed for flaviviruses such as Bamaga virus [26]. This probably resulted from the varied midgut barrier expressed following ingestion of the virus in different mosquito species [27]. This as well suggested that *Cx. quinquefasciatus* and *Ae. albopictus* express a midgut escape barrier, as none of the infected *Ae. albopictus* and *Cx. quinquefasciatus* showed a disseminated infection. A previous study showed that Eilat virus (EILV, an ISV) could bypass midgut barriers and spread to the salivary glands of *Ae. albopictus* and *Cx. quinquefasciatus* treated with thoracic injection [23]. We attempted to address the ability of *Cx. quinquefasciatus* and *Ae. albopictus* to acquire viruses from water during the larval stage. Several studies have shown that the population of adult *Ae. aegypti* mosquitoes could be effectively infected by the virus in sewage and clean water [28]. Accumulatively, evidence from field surveillance convincingly shows that *A. aegypti* and *A. albopictus* tend to oviposit and breed in wastewater with low dissolved oxygen and high turbidity [29]. Our study further confirmed that YCV in an aquatic environment can be acquired and subsequently transmitted by mosquitoes. Interestingly, the infectivity of YCV in sewage is higher than that in clean water, and certain ingredients in the sewage can affect viral infections. Indeed, as shown in S1 Table, we noted that the aquatic particles correlated with YCV infectivity. The difference might be caused by the divalent cations or the pH, as suggested in other studies on *Culex restuans Cypovirus* (CrCPV), a dsRNA virus [30,31],

*Uranotaenia sapphirina Nucleopolyhedrovirus* (UrsaNPV)[32], a DNA virus and ZIKV [29], or perhaps the presence of bacteria, as demonstrated in Reoviruses by other authors [33]. The effects of divalent cations or pH on YCV infectivity in water however remains to be further investigated.

Although vertical transmission has been considered an important way for ISVs to persist and disperse in nature, the relatively low proportion of YCV positive F1 generation (eggs, larvae or adult) corroborates low rates observed for *Culex flavivirus*, *Aedes flavivirus*, *Okushiri virus*, and *Kamiti River virus* [9,34–42]. It is possible that ISVs might persist in mosquitoes for a long time, although the vertical transmission efficiency may be low. Indeed, studies of *Dipteran ambidensovirus* 1 (*Cx. pipiens densovirus*) proved that the virus can persist for over 20 years in laboratory colony *Cx. pipiens* (sl.), with a low rates of vertical and transovarial transmission [43]. The low rate of YCV vertical transmission was consistent with the midgut restriction barrier expressed in the mosquito. Indeed, the virus disseminated from the midgut cells typically undergoes secondary replication in other tissues, such as fat bodies or ovaries [27]. Collectively, the vertical transmission capacity of YCV is low and horizontal transmission is also restricted to a certain extent. Mixed-mode transmission, including both horizontal and vertical transmission routes, is likely to be key for YCV to be maintained in nature, which is similar to ISFs [44,45].

Superinfection exclusion is an important feature of ISVs. There are several studies of superinfection interference, which mostly occur between cognate viruses [13,46,47], as the mechanism is generally considered to be competitive inhibition. In addition, the interference could be caused by indirectly regulate the immune system of the host. *Mesoniviridae* is far from the *Flaviviridae*, and their interactions are rarely reported. Several reports suggest that flaviviruses do not interfere with viruses belonging to other families or genera in most instances [48,49]. However, other authors report conflicting results [50,51], which seem to agree with our data. A recent publication [52] demonstrated that EILV induces heterologous interference with several other *Alphavirus* pathogens, suggesting that heterologous interactions can occur. Indeed, the data in our laboratory reveal that the inhibitory effect of YCV on flaviviruses was also different, since the most pronounced reduction was observed in sequential infection with DENV-2 ($10^4$-fold less), whereas in JEV, no reduction was seen in coinfections or sequential infections. This could be the critical factors directly or indirectly affected by the YCV of these medically important flaviviruses that are viral species specific, and further study such as direct RNA sequencing for the viruses or transcriptome analysis for the host cells could help to explain this difference. Even though the prevalence of arthropod-borne viruses in mosquito populations is quite low, the transmission of mosquito bites would greatly increase the epidemic risk of arbovirus-related diseases. Therefore, research on the interaction between microorganisms in mosquitoes is of utmost importance for epidemiological prediction and biological control of arboviruses.

ISVs are only maintained in insect populations, however it remains unclear how they transmit among insects. Here, we report that YCV could infect mosquitoes orally at the adult stage and breed in aquatic environments at the larva stage, with a low vertical transmission capacity. This offers laboratory evidence that mixed-mode transmission in mosquitoes, including both horizontal and vertical transmission routes, is likely to be the key for YCV maintenance in nature. The growth and tissue distribution of YCV in mosquitoes suggests that YCV cannot spread to the host salivary glands and that *Ae. albopictus* maybe a better host for YCV than *Cx. quinquefasciatus*. Interactive data between YCV and three flaviviruses indicate that superinfection exclusion may occur not only between homologous viruses, but also between heterologous viruses, and the inhibition does not occur with all flaviviruses. It is equally important to understand the ecological significance of this interaction to expedite the understanding

mechanisms of interference between vector-borne viruses and ISVs in mosquitoes and subsequent implementation of vector-borne virus control.

## Supporting information

**S1 Fig. *Cx. quinquefasciatus* and *Ae. albopictus* are permissive to YCV infection when fed a blood meal.** Mosquitoes were inoculated with an infectious blood meal containing a dose of $10^6$ pfu/ml YCV. Mosquitoes were homogenized and YCV reproduction was detected by reverse transcription PCR (A) and Western blot (B) at 14 dpi. Homogenate of mosquitoes at 14 dpi was added to C6/36 cells, infectious virus particles in the whole mosquito bodies were assessed by the cytopathic effect (CPE) (C) and viruses in the supernatant of C6/36 cells were detected by reverse transcription PCR (D).
(TIF)

**S2 Fig. YCV effects the growth of DENV-2, ZIKV and JEV in C6/36 and Aag2 cells during single- and coinfections under simultaneous infections.** The flaviviruses DENV-2, ZIKV and JEV at MOI 1 were mixed with YCV at MOI 1 or 0.1; then, the mixture was added to C6/36 (left) or Aag2 (right) cells. (A, B, C) The virus titers during single- and coinfections were determined by the plaque assay at the indicated time points. Data are presented as the mean of three independent experiments ±SEM. The results were analyzed using the unpaired t-test. A P value of $< 0.05$ indicates statistical significance. $P < 0.05$, *; $P < 0.001$, ** and $P < 0.001$, ***.
(TIF)

**S3 Fig. YCV effects the growth of DENV-2, ZIKV and JEV in C6/36 cells during single- and coinfections under sequential infections.** After 12 h of YCV (MOI 1) infection, C6/36 cells were infected with the flaviviruses DENV-2, ZIKV or JEV at MOI 1. (A, B, C) The virus titers during single- and coinfection were determined by the plaque assay at the indicated time points. Data are presented as the mean of three independent experiments ±SEM. The results were analyzed using unpaired t-test. A P value of $< 0.05$ indicates statistical significance. $P < 0.05$, *; $P < 0.001$, ** and $P < 0.001$, ***.
(TIF)

**S1 Table. Characterization of the clean water and sewage.**
(DOCX)

## Acknowledgments

We are thankful to Bo Zhang, Wuhan Institute of Virology, Chinese Academy of Sciences, for kindly provided DENV-2 (strain TSV01), ZIKV strain (SZ-WIV01) and JEV (strain SA14). And we would like to thank the Animal Center, Wuhan Institute of Virology, Chinese Academy of Science, for technical assistance.

## Author Contributions

**Data curation:** Guoguo Ye, Qiannan Dong.

**Formal analysis:** Guoguo Ye.

**Funding acquisition:** Zhiming Yuan, Han Xia.

**Methodology:** Qiannan Dong, Quanxin Cai.

**Resources:** Quanxin Cai.

**Supervision:** Zhiming Yuan, Han Xia.

**Writing – original draft:** Guoguo Ye.

**Writing – review & editing:** Yujuan Wang, Xiaoyun Liu, Zhiming Yuan, Han Xia.

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
