## [Decision Letter · Decision Letter 0]

28 Jul 2020

Dear Dr. Xia,

Thank you very much for submitting your manuscript "Transmission competence of a new mesonivirus, Yichang virus, in mosquitoes and its interference with representative flaviviruses in vitro" for consideration at PLOS Neglected Tropical Diseases. As with all papers reviewed by the journal, your manuscript was reviewed by members of the editorial board and by several independent reviewers. In light of the reviews (below this email), we would like to invite the resubmission of a significantly-revised version that takes into account the reviewers' comments. 

Major editorial issues have been raised by the editors, as well as those touching on interpretation and limitations of the study that must be addressed by the authors before the manuscript can be further considered for publication.

We cannot make any decision about publication until we have seen the revised manuscript and your response to the reviewers' comments. Your revised manuscript is also likely to be sent to reviewers for further evaluation.

Sincerely,

Paul O. Mireji, PhD

Associate Editor

A. Desiree LaBeaud

Deputy Editor

Major editorial issues have been raised by the editors, as well as those touching on interpretation and limitations of the study that must be addressed by the authors before the manuscript can be further considered for publication.

Reviewer's Responses to Questions

**Key Review Criteria Required for Acceptance?**

**Methods**

-Are the objectives of the study clearly articulated with a clear testable hypothesis stated?

-Is the study design appropriate to address the stated objectives?

-Is the population clearly described and appropriate for the hypothesis being tested?

-Is the sample size sufficient to ensure adequate power to address the hypothesis being tested?

-Were correct statistical analysis used to support conclusions?

-Are there concerns about ethical or regulatory requirements being met?

Reviewer #1: (No Response)

Reviewer #2: -Are the objectives of the study clearly articulated with a clear testable hypothesis stated?

yes the objectives were clearly articulated

-Is the study design appropriate to address the stated objectives?

study design was appropriate to answer the objectives

-Is the population clearly described and appropriate for the hypothesis being tested?

yes 

-Is the sample size sufficient to ensure adequate power to address the hypothesis being tested?

yes 

-Were correct statistical analysis used to support conclusions?

yes

-Are there concerns about ethical or regulatory requirements being met?

yes

Reviewer #3: Sample sizes and replicates of experiments are not clearly stated. More detail on the experiments conducted and what each tested need to be provided, Some methods details are misplaced in the results section

**Results**

-Does the analysis presented match the analysis plan?

-Are the results clearly and completely presented?

-Are the figures (Tables, Images) of sufficient quality for clarity?

Reviewer #1: The author of this manuscript has isolated the virus,Yichang virus, YCV of Mesoniviridae from mosquitoes in China. In this manuscript, the researchers conducted a viral infection test on adult mosquitoes and larvae of Culex quinquefasciatus and Aedes albopictus, and found that YCV can be detected in the midgut of mosquitoes, YCV has limited vertical transmission in mosquitoes, and YCV can replicate in adults of Culex quinquefasciatus and Aedes albopictus.

Reviewer #2: -Does the analysis presented match the analysis plan?

yes it does 

-Are the results clearly and completely presented?

results are clearly and completely presented 

-Are the figures (Tables, Images) of sufficient quality for clarity?

yes figures were of good quality

Reviewer #3: The results section also includes methods and discussion of the results, which should be moved to the appropriate sections. The results could be condensed by referring more to the figures regarding key finding, but not describing all of the results in each of the figures in such detail.

**Conclusions**

-Are the conclusions supported by the data presented?

-Are the limitations of analysis clearly described?

-Do the authors discuss how these data can be helpful to advance our understanding of the topic under study?

-Is public health relevance addressed?

Reviewer #1: Although the results of this study provide some biological characteristics of YCV in mosquitoes (Culex quinquefasciatus and Aedes albopictus), this manuscript only observes the research data on the co-infection of YCV and Flavivirus (Dengue virus, Zika virus, Japanese encephalitis virus) in mosquito cell lines (C6/36). This research lacks the infection results of YCV and the above Flavivirus which takes adult mosquitoes as the research object, so the research results are not enough to support the conclusion that YCV has replication interference effect on Flavivirus.

Reviewer #2: -Are the conclusions supported by the data presented?

the dots presented supports the conclusion

-Are the limitations of analysis clearly described?

none 

-Do the authors discuss how these data can be helpful to advance our understanding of the topic under study?

yes the data is nicely presented , but can be further enriched if they can extend their discussions and thoughts on why JEV was not affected by YCV

-Is public health relevance addressed?

yes it is of relevance to public health

Reviewer #3: I believe that they are although I do point out some overstatements in the attached track-changes document

**Editorial and Data Presentation Modifications?**

Reviewer #1: (No Response)

Reviewer #2: The manuscript was well written and presented and advances our understanding on microbial interactions and arbovirus transmission. There was only a minor error as highlighted on lines 34-36. the authors wrote YCV is more susceptible to its hosts rather than the host being more susceptible to the virus. 

The other comment is on line 320 DKV is mentioned for the first time. its general to give the full name then the abbreviation later for ease of reading by audience who may not be familiar with the field.

Reviewer #3: In the attached track-changes version, I have made a number of grammatical and tense edits. I also left a few comments.

**Summary and General Comments**

Reviewer #1: It is recommended to submit the manuscript after supplementing the test results.

Reviewer #2: in general the manuscript is well written and presented and provides new and exciting data to arbovirus research and the general scientific audience. The minor comments raised can easily be addressed and manuscript accepted for publication

Reviewer #3: This is a very interesting study showing that Yichang virus may interfere with flavivirus transmission in mosquitoes and should be published after clarifications.

PLOS authors have the option to publish the peer review history of their article (what does this mean?). If published, this will include your full peer review and any attached files.

Reviewer #1: No

Reviewer #2: Yes: Martin Rono

Reviewer #3: No
---

## [Decision Letter · Decision Letter 1]

16 Oct 2020

Dear Dr. Xia,

We are pleased to inform you that your manuscript 'Transmission competence of a new mesonivirus, Yichang virus, in mosquitoes and its interference with representative flaviviruses' has been provisionally accepted for publication in PLOS Neglected Tropical Diseases.

Best regards,

Paul O. Mireji, PhD

Associate Editor

A. Desiree LaBeaud

Deputy Editor

Reviewer's Responses to Questions

**Key Review Criteria Required for Acceptance?**

**Methods**

-Are the objectives of the study clearly articulated with a clear testable hypothesis stated?

-Is the study design appropriate to address the stated objectives?

-Is the population clearly described and appropriate for the hypothesis being tested?

-Is the sample size sufficient to ensure adequate power to address the hypothesis being tested?

-Were correct statistical analysis used to support conclusions?

-Are there concerns about ethical or regulatory requirements being met?

Reviewer #1: The study is clearly with the hypothese and acceptable to publicaton

Reviewer #2: All the issues previously raised have been addressed. Accept the manuscript

**Results**

-Does the analysis presented match the analysis plan?

-Are the results clearly and completely presented?

-Are the figures (Tables, Images) of sufficient quality for clarity?

Reviewer #1: The results have been completely presented.

Reviewer #2: the analysis presented match the analytical plan and results are well presented. The figures and table are of acceptable quality

**Conclusions**

-Are the conclusions supported by the data presented?

-Are the limitations of analysis clearly described?

-Do the authors discuss how these data can be helpful to advance our understanding of the topic under study?

-Is public health relevance addressed?

Reviewer #1: The conclusion has been supported by the data presented.

Reviewer #2: the conclusions are supported by the data presented

**Editorial and Data Presentation Modifications?**

Reviewer #1: Accept

Reviewer #2: Accept

**Summary and General Comments**

Reviewer #1: Accepted

Reviewer #2: The revised manuscript has addressed previous issues raised by reviewers. The study presents new data and insights about Yichang virus potential role for biological control for DENV-2 transmission by Ae. albopictus mosquitos.

PLOS authors have the option to publish the peer review history of their article (what does this mean?). If published, this will include your full peer review and any attached files.

Reviewer #1: No

Reviewer #2: No

---

## [Editor Report · Acceptance letter]

17 Nov 2020

Dear Dr. Xia,

We are delighted to inform you that your manuscript, "Transmission competence of a new mesonivirus, Yichang virus, in mosquitoes and its interference with representative flaviviruses," has been formally accepted for publication in PLOS Neglected Tropical Diseases.

Best regards,

Shaden Kamhawi

co-Editor-in-Chief

Paul Brindley

co-Editor-in-Chief
